# Multimodal Sentiment Analysis To Explore the Structure of Emotions

## Abstract

We propose a novel approach to multimodal sentiment analysis using deep neural networks combining visual recognition and natural language processing. Our goal is different than the standard sentiment analysis goal of predicting whether a sentence expresses positive or negative sentiment; instead, we aim to infer the latent emotional state of the user. Thus, we focus on predicting the emotion word tags attached by users to their Tumblr posts, treating these as "self-reported emotions." We demonstrate that our multimodal model combining both text and image features outperforms separate models based solely on either images or text. Our model's results are interpretable, automatically yielding sensible word lists associated with emotions. We explore the structure of emotions implied by our model and compare it to what has been posited in the psychology literature, and validate our model on a set of images that have been used in psychology studies. Finally, our work also provides a useful tool for the growing academic study of images—both photographs and memes—on social networks.

## 1 Introduction

Sentiment analysis has been an active area of research in the past decade, especially on textual data from Twitter, e.g. early work by Pak & Paroubek (2010) showed that emoticons could be used to collect a labeled dataset for sentiment analysis, Golder & Macy (2011) investigated temporal patterns in emotion using tweets, and Bollen et al. (2011) investigated the impact of collective mood states on the stock market. The SemEval series of "Sentiment Analysis in Twitter" challenges has used Twitter data as a benchmark to spur the development of new sentiment analysis algorithms (Rosenthal et al., 2017).

Unlike Twitter, Tumblr posts are not limited to 140 characters, allowing more expressiveness, and they often focus on visual content: most Tumblr posts contain an image with some accompanying text. As pictures—both photographs and memes—have become prevalent on social media, researchers have begun to study them, and make novel claims about the role that they play in social media. Shifman (2014) makes an argument for taking memes seriously, and Miller & Sinanan (2017) use memes throughout their cross-country anthropological study of Facebook, characterizing who posts what sorts of memes and what sort of communicative function memes play. Inspired by this research, we take images seriously as a source of data worth analyzing. Further, we aim to enable a research agenda focused on images by giving social scientists tools to address fundamental questions about the use of images on social media.

In psychology, the gold standard for measuring emotions is self-report, i.e. if an individual says that they are happy then that is taken to be the truth (Gilbert, 2006). On Tumblr, users often attach tags to their posts which we consider to be emotional self-reports, as these tags take the simple form of, e.g. "#happy". By collecting a large dataset and using these emotion word tags as labels, we argue that our sentiment analysis approach, which combines images and text, leads to a more psychologically plausible model, as the dataset combines two rich sources of information and has labels we believe are a good proxy for self-reported emotion.

Concretely, the Deep Sentiment model associates the features learned by the two modalities as follows:

- For images, we fine-tune Inception (Szegedy et al., 2015), a pre-trained deep convolutional neural network, to our specific task of emotion inferring.

- The text is mapped into a rich high-dimensional space using a word representation learned by GloVe (Pennington et al., 2014). The embedded vectors are then fed to a recurrent network which preserves the word order and captures some of the semantics of human language.

- A dense layer combines the information in the two modalities and a final softmax output layer gives the probability distribution over the possible emotion word tags.

## 2  RELATED WORK

Visual sentiment analysis has received much less attention compared to text-based sentiment analysis. Yet, images are a valuable source of information for accurately inferring emotional states as they have become ubiquitous on social media as a means for users to express themselves (Miller & Sinanan, 2017). Although huge progress has been made on standard image classification tasks thanks to the ImageNet challenge (Russakovsky et al., 2015), visual sentiment analysis may be fundamentally different from classifying images as it requires a higher level of abstraction to understand the message conveyed by an image (Joshi et al., 2011). Implicit knowledge linked to culture and intrinsic human subjectivity – that can make two people use the same image to express different emotions – makes visual sentiment analysis a difficult task.

Borth et al. (2013) pioneered sentiment analysis on visual content with SentiBank, a system extracting mid-level semantic attributes from images. These semantic features are outputs of classifiers that can predict the relevance of an image with regard to one of the emotions in the Plutchik's wheel of emotions (Plutchik, 2001). Motivated by the progress of deep learning methods, You et al. (2015) used convolutional neural networks on Flickr with domain transfer from Twitter for binary sentiment classification. However, studies about image annotation showed that combining text features with images can greatly improve performance as shown by Guillaumin et al. (2010) and Gong et al. (2014).

Successful results in multimodal sentiment analysis have been achieved using non-negative matrix factorisation (Wang et al., 2015) and latent correlations (Katsurai & Satoh, 2016). Chen et al. (2015) investigated the image posting behaviour of social media users and found in his study that two thirds of the participants added an image to their tweets to enhance the emotion of the text. In order to uncover the link between image tweets and the latent emotion, he used Latent Dirichlet Allocation to model image tweets with three modalities: textual, visual and emotional.

One weakness shared by the papers above is the lack of large training datasets due to the inherent complexity of finding labeled images/text on social media. In this work, we aim to mitigate this problem using a large noisy labeled dataset of Tumblr posts. Further, sentiment analysis on text is a well-developed research area in both computer science and psychology, and sentiment analysis has been used to answer psychological questions. However, researchers have cautioned that sentiment analysis focuses on the positive or negative sentiment expressed by a piece of text, rather than on the underlying emotional state of the person who wrote the text (Flaxman & Kassam, 2016) and thus is not necessarily a reliable measure of latent emotion. We address this problem by trying to predict emotional state of the user instead of the sentiment polarity.

## 3  TUMBLR DATASET

Tumblr is a microblogging service where users post multimedia content that often contains the following attributes: an image, text, and tags. The critical piece of our approach, which distinguishes it from sentiment analysis methods focused on purely distinguishing positive from negative, is that we use the emotion word tags as the labels we wish to predict. As we consider these emotion word tags to be (noisy) labels indicating the user's state of mind when writing a post, we can use them as a proxy for self-reported emotion, and thus a proxy for the underlying emotional state of the user.

To build our dataset, queries were made through the Tumblr API searching for an emotion appearing in the tags. The 15 emotions retained were those with high relative frequencies on Tumblr among the PANAS-X scale (Watson & Clark, 1999) or the Plutchik's wheel of emotions (Plutchik, 2001).

In some posts, the tag containing the emotion of the post also appeared in the text itself. We removed these words from the text in order not to give our classifier an unfair advantage. We extracted every tagged posts from Tumblr by going backward from 2017 until 2011. Due to the API limitations, posts from high density emotions, such as 'happy' or 'angry', did not go as far back in the past, but otherwise we believe that our dataset is a complete sample.

As Tumblr is used worldwide, we had to filter out non-English posts: posts with less than 50% of English words were removed from the dataset. The vocabulary of English words was obtained from GloVe (although GloVe's vocabulary contains some non-English words this approach filtered out non-English posts reasonably well). Also, not every extracted post contained an image and we likewise excluded these. Figure 1 shows two posts with their associated emotions (more examples in Appendix A) and Table 1 summarises the statistics of the data[1].

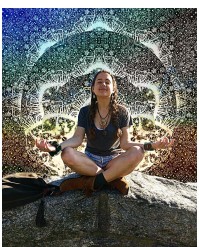 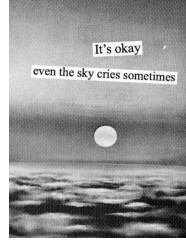

(a) **Optimistic**: "The most beautiful thing we can experience is the mysterious. It is the source of all true art and science – Albert Einstein."

(b) **Sad**: "It's okay to be upset. It's okay to not always be happy. It's okay to cry. Never hide your emotions in fear of upsetting others or of being a bother."

Figure 1: Examples of Tumblr posts

Table 1: Summary statistics for the Tumblr dataset, with posts from January 2011 to September 2017.

**Tumblr data**

| Posts | English text | English text and image |
|---|---|---|
| 1,009,534 | 578,699 | 256,897 |

| Emotion | Posts | English text | English text and image |
|---|---|---|---|
| **Happy** | 189,841 | 62% | 29% |
| **Calm** | 139,911 | 37% | 29% |
| **Sad** | 124,900 | 53% | 15% |
| **Scared** | 104,161 | 65% | 20% |
| **Bored** | 101,856 | 54% | 29% |
| **Angry** | 100,033 | 60% | 21% |
| **Annoyed** | 72,993 | 78% | 10% |
| **Love** | 66,146 | 61% | 39% |
| **Excited** | 37,240 | 58% | 41% |
| **Surprised** | 18,322 | 47% | 32% |
| **Optimistic** | 16,111 | 64% | 36% |
| **Amazed** | 10,367 | 61% | 35% |
| **Ashamed** | 10,066 | 63% | 22% |
| **Disgusted** | 9,178 | 69% | 17% |
| **Pensive** | 8,409 | 57% | 34% |

---

[1]We cannot redistribute our dataset due to licensing restrictions, but the code to replicate the dataset and the results is available on: `https://github.com/deepsentiment/deepsentiment`

# 4 METHODOLOGY

## 4.1 VISUAL ANALYSIS

Training a convolutional network from scratch can be challenging as a large amount of data is needed and many different architectures have to be tried before achieving satisfying performances. To circumvent this issue, we can take advantage of a pre-trained network named Inception (Szegedy et al., 2015) that learned to recognise images through the ImageNet dataset with a deep architecture of 22 layers.

Inception learned representations capturing the colors and arrangement of shapes of an image, which turn out to be relevant when dealing with images even for a different task. We could also say that the pre-trained network grasped the underlying structure of images. This statement rests on the hypothesis that all images lie in a low-dimensional manifold, and recent advances in realistic photos generation through generative adversarial networks bolsters this idea (Radford et al., 2016).

## 4.2 NATURAL LANGUAGE PROCESSING

Even as a human being, it can be difficult to guess the expressed emotion only by looking at a Tumblr image without reading its caption as shown by Figure 2.

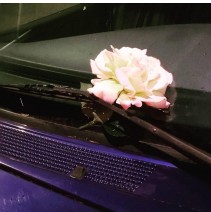

Figure 2: Which emotion is it?

It is unclear whether the user wants to convey happiness or surprise. Only after reading the accompanying text, "To whoever left this on my windshield outside of last night's art opening, I love you. You made my night," can we finally conclude that the person was *surprised* (and possibly also feeling other emotions like amazed). The text is extremely informative and is usually crucial to accurately infer the emotional state.

### 4.2.1 WORD EMBEDDING

Most learning algorithms rely on the local smoothness hypothesis, that is, similar training instances are spatially close. This hypothesis clearly doesn't hold with words one-hot encoded as for instance 'dog' is as close to 'tree' as it is to 'cat'. Ideally, we would like to transform the word 'dog' into a space in which it is closer to 'cat' than it is to 'tree'. Word embeddings produce a mapping in which words are projected into a high-dimensional space that preserves semantic relationships. We use the GloVe (Pennington et al., 2014) word embedding that was trained on Twitter data, because the writing styles on Twitter and Tumblr are similar.

Each post in the dataset does not necessarily contain the same number of words. Even after embedding each word, the input will be of variable size and most learning algorithm expect a fixed-sized input. We could simply average across the high-dimensional representations of the words and take a distribution regression approach (Muandet et al., 2017). However note that averaging would mean that the word order would be completely lost. Human language relies heavily on word order to communicate as for example the word *change* can be both a noun and a verb, and negation such as 'not entertained' can only be understood if 'not' directly precedes the verb. We will preserve word order by using recurrent neural networks.

### 4.2.2 Sequence input

Models of natural language using neural networks have proved to outperform the more traditional statistical models that were limited by the Markov assumption (Bengio et al., 2003; Goodman, 2001). One explanation could be that the compact representation of words through word embeddings is robust (Mikolov et al., 2011) and do not need any smoothing over probabilities. Among the neural models, the recurrent-based models allows for short-term memory inspired by how humans read sentences: past context is essential to understand the meaning of written language. Contrary to shallow feedforward networks, that can only cluster similar words, recurrent networks (which can be viewed as a deep architecture (Bengio & LeCun, 2007)) can cluster similar histories. Recurrent neural networks have a temporal awareness represented by the hidden state that can be seen as an embedding of the past words. For example in Sutskever et al. (2014) a quality translation of a sentence was made possible with the last output of a recurrent neural network.

In our setting, for a given Tumblr post, the text is broken down into a sequence of words that are embedded into a high-dimensional space (unknown words are mapped to the zero vector) and then fed into an LSTM (Hochreiter & Schmidhuber, 1997). To account for shorter posts, we zero-pad the vector with a special word token. For longer posts, we only keep the 50 first words which is a reasonable choice as 76% posts in the dataset contain less than 50 words.

### 4.3 Deep Sentiment: a multimodal neural network

Information often comes in several modalities, and humans are able to seamlessly combine them. For instance, in speech recognition, humans integrate audio and visual information to understand speech, as was demonstrated by the McGurk effect (McGurk & MacDonald, 1976). Separating what we see from what we hear seems like an easy task, but in an experiment conducted by McGurk, the subjects who were listening to a /ba/ sound with a visual /ga/ actually reported they were hearing a /da/. This is uncanny as even if we know the actual sound is a /ba/, we cannot stop our brain from interpreting it as a /da/.

In Figure 2 we gave an example of text being necessary to fully understand the emotion expressed by an image. Sometimes alternative text would lead to entirely different interpretations, as shown in Figure 3:

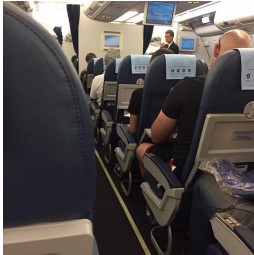
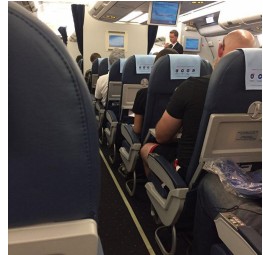

(a) **Scared**: "Planes might just be the most frightening thing ever."

(b) **Angry**: "I hate it when people are taking too much space on planes."

Figure 3: Different meanings with different captions.

Exploiting both visual and textual information is therefore key to understanding a user's underlying emotional state. We call our proposed network architecture, combining visual recognition and text analysis, "Deep Sentiment".

### 4.3.1 Architecture

Deep Sentiment's architecture is shown in Figure 4:

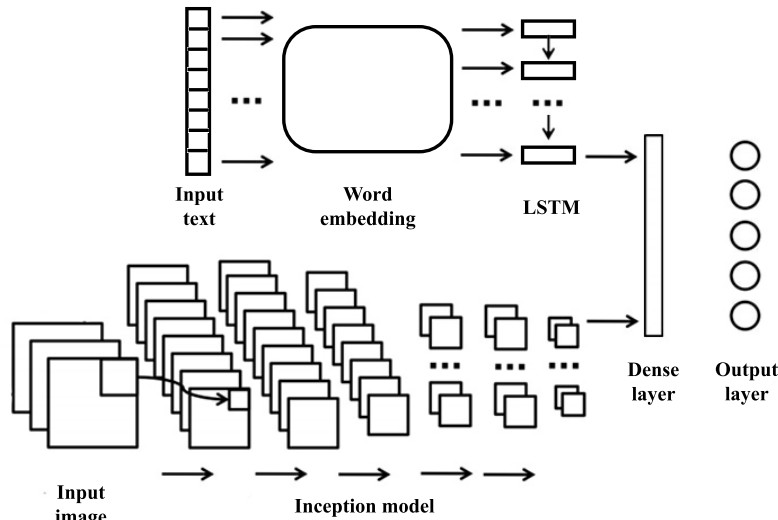

Figure 4: The Deep Sentiment structure. On the one hand, the input image, resized to (224,224,3) is fed into the Inception network and outputs a vector of size 256. On the other hand, the text is projected into a high-dimensional space that subsequently goes through an LSTM layer with 1024 units. The two modalities are then concatenated and fed into a dense layer. The final softmax output layer give the probability distribution over the emotional state of the user.

## 5 EVALUATION

In Table 2, we compare Deep Sentiment with the image model (Inception model fine-tuned through the last Inception module), the text model, and a baseline: random guessing that includes the prior probabilities of the classes. Figure 5 shows a comparison of the accuracy curves of the different models.

Table 2: Comparison of image model, text model and Deep Sentiment.

|  | Loss | Train accuracy | Test accuracy |
|---|---|---|---|
| **Random guessing** | - | 11% | 11% |
| **Image model** | 1.80 | 43% | 36% |
| **Text model** | 0.81 | 72% | 69% |
| **Deep Sentiment** | 0.75 | 80% | 72% |

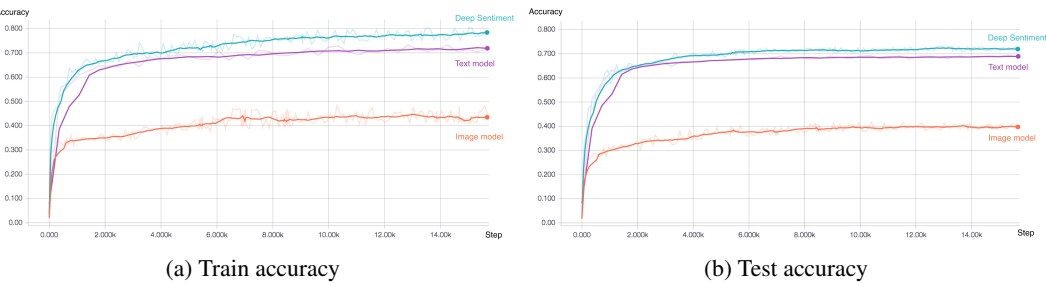

(a) Train accuracy          (b) Test accuracy

Figure 5: Accuracy comparison of the three models

Using text alone, the test accuracy is 69%, almost double the accuracy of the image model, this suggests that on Tumblr, text is a better predictor of emotion than images, as we illustrated in Figure 2. By combining text and images Deep Sentiment achieves 80% train accuracy and 72% test accuracy, significantly outperforming the images-only model and slightly outperforming the text-only model (note that no validation set was used to tune hyperparameters, even though various architectures were tried before yielding Deep Sentiment).

## 6 RESULTS

In this section, we carefully investigate what psychologically meaningful results we can draw from our model, and whether they match previous results in the psychology literature.

### 6.1 TOP WORDS FOR EACH EMOTION

We investigated which words were the most relevant for each emotion as follows: we created artificial posts whose text consisted of a single word (each of the most frequent 1,000 words in the whole dataset were tested) accompanied by an image that was the mean image (per channel mean). For each emotion, we report the 10 highest scoring words from these 1,000 artificial posts in Table 3 (words 11-20 for each emotion are in Appendix B).

Table 3: Top 10 words for each emotion, ordered by the relative frequency of the emotion being used as a tag on Tumblr

| Emotion | Top words |
|---|---|
| **Happy** | healthy, loving, enjoy, wonderful, warm, happiness, smile, lovely, cute, proud |
| **Calm** | quiet, situation, peace, mood, towards, warm, slowly, stay, sleep, rain |
| **Sad** | horrible, sorry, crying, hurts, tears, cried, lonely, memories, worst, pain |
| **Scared** | terrified, scary, panic, nervous, fear, afraid, horrible, woke, happening, worried |
| **Bored** | asleep, tired, kinda, busy, stuck, constantly, lonely, sat, listening, depression |
| **Angry** | anger, fear, panic, annoying, hate, mad, upset, anxiety, scares, stupid |
| **Annoyed** | pissed, ashamed, angry, nervous, speak, surprised, tired, worried, ignore, phone |
| **Love** | soul, dreams, happiness, kiss, sex, beauty, women, feelings, god, relationships |
| **Excited** | tonight, hopefully, watching, nervous, surprised, expect, tomorrow, amazing, hoping, happen |
| **Surprised** | birthday, cried, thank, yesterday, told, sorry, amazing, sweet, friend, message |
| **Optimistic** | positive, expect, surprised, healthy, grow, realize, clearly, hopefully, calm, peace |
| **Amazed** | surprised, excited, amazing, woke, realized, awesome, happening, ashamed, yeah, happened |
| **Ashamed** | totally, honestly, sorry, absolutely, freaking, honest, completely, stupid, seriously, am |
| **Disgusted** | ashamed, totally, angry, hate, stupid, annoyed, horrible, scares, freaking, absolutely |
| **Pensive** | mood, wrote, quiet, view, sadness, thoughts, calm, words, sad, kissed |

An inspection of each of the words suggests that none are out of place, with the possible exception of "ashamed" in the list of Amazed words. (This may be due to the fact that the tag Amazed is sometimes used ironically.) Further, our data-driven approach suggests that our methods could be used as an alternative to the word list-based approaches to sentiment analysis common in psychology. The most widely used tool, Linguistic Inquiry Word Count (LIWC), (Pennebaker et al., 2007), consists of dozens of English words which were compiled by hand into psychologically meaningful categories such as "Health/illness" or "Anxiety" and used in a large number of psychology studies (Tausczik & Pennebaker, 2010). Not only does our approach automatically give sensible word lists, it contains modern words, common in social media usage, like "woke" (first attested in its modern meaning in 1962 according to the Oxford English Dictionary which defines it as "alert to racial or social discrimination and injustice") and "phone." Woke appears in the top 10 of "Scared" and "Amazed" and in the top 20 of "Calm" (Appendix B), an interesting finding in its own right. By contrast, both woke and phone appear in LIWC, but not in any of LIWC's emotion word categories, only in a list of verbs and a list of social words, respectively.

## 6.2 CLUSTERING EMOTIONS

Psychologists have long studied the structure of emotion, and debated whether there are a small number of "core" emotions (Ekman, 1992), two dominant factors (see Tellegen et al. (1999) for a discussion of various theories), or more complex models (e.g. Lindquist et al. (2013) critiques previous models and argues that emotions do not belong in natural categories or along multiple dimensions, because they are the "constructions" of the human mind, and the result of complex perceptions). Classical attempts by psychologists to answer this question have relied on survey data in which respondents can self-report more than one emotion, followed by clustering or factor analysis approaches. More recently, psychologists have turned to neuroscience to address these questions.

While we collected a dataset in which each Tumblr post was labeled with a single emotion tag, we can nevertheless directly address this question using our trained model. Given an image $\mathcal{I}$ and text $\mathcal{T}$ from our dataset, we can compute the conditional probabilities of the emotion classes: $\hat{y}|\mathcal{I}, \mathcal{T}$ where $\hat{y}$ is a probability vector in the 15-dimensional simplex. To calculate our model's predicted empirical distribution over $\hat{y}$ we predict $\hat{y}$ for all posts in our dataset: $(\hat{y}_i|\mathcal{I}_i, \mathcal{T}_i)_{i=1}^n$. Finally, we calculate the empirical correlation matrix of $\hat{y}_1, \ldots, \hat{y}_n$, which we include in the Appendix in Figure 9. For ease of visualization, we convert the correlation matrix to a distance matrix and perform hierarchical clustering as shown in Figure 6.

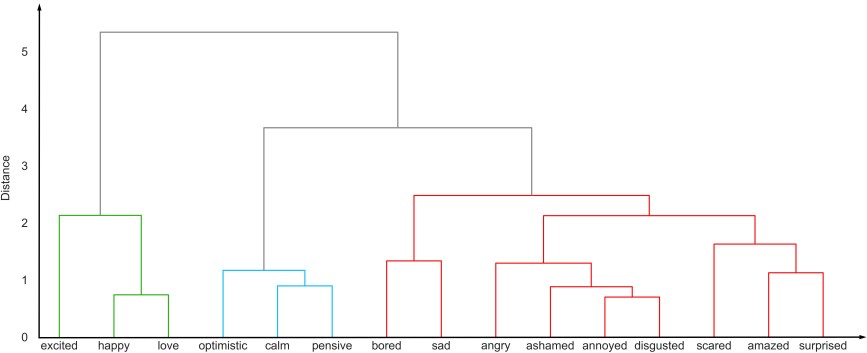

Figure 6: Hierarchical clustering of the emotion correlation matrix.

Similar to almost all previous psychology studies, there is a clear distinction between emotions that are **positive** in nature, such as excited, happy, and love and emotions that are **negative** such as angry, ashamed, annoyed, and disgusted. Another finding consistent with much previous literature, which we investigate more below, is that within both positive and negative emotions, there is a distinction between low arousal emotions and high arousal emotions. Bored and sad (low arousal) are in one cluster, while high arousal emotions—angry, ashamed, annoyed, disgusted—are in another.

## 6.3 THE CIRCUMPLEX MODEL OF EMOTION

The circumplex model (Posner et al., 2005) posits two dimensions which explain emotions, usually valence (positive vs. negative) and arousal (low vs. high), though other dimensions have been suggested. We investigated whether two factors explained most of the variance in our results using principal component analysis (PCA). As shown in the scree plot in the Appendix in Figure 12, while the first component explains 28.8% of the variance, dimensions 2 and 3 explain 16.4% and 14.6% respectively, evidence against a simple two factor model.

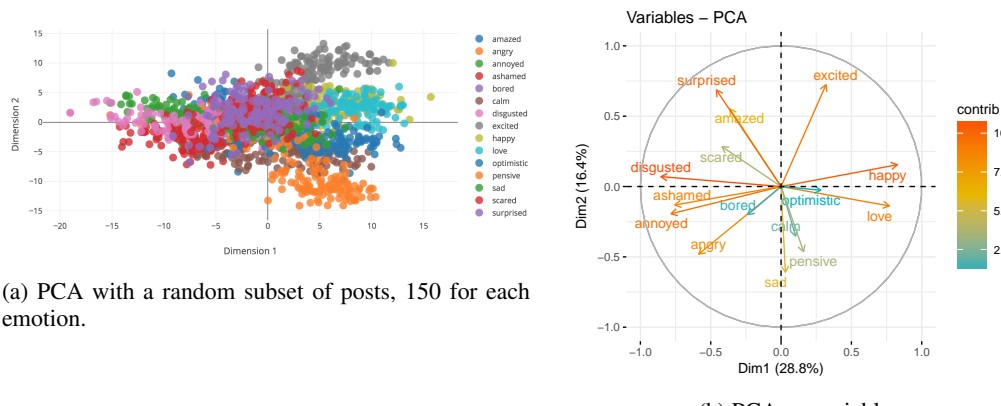

(a) PCA with a random subset of posts, 150 for each emotion.

(b) PCA on variables

Figure 7: Affect/valence

We visualize the results of PCA in Figure 7. At left, a random subset of posts, where for visualization purpose we sampled equally from each emotion class, are shown along the first two principal components, and each is colored by the most probable predicted emotion. This shows that within any particular emotion category there is much variation along these two dimensions, i.e. they do not cluster very tightly. Related emotions like happy and love are close together, as they were in the dendrogram.

At right, we visualise the projection of the emotion variables onto the first two principal components. Using this figure, we conclude that the first principal component corresponds well to valence, as it neatly separates happy/love from disgusted/ashamed/annoyed. However, the second principal component is not clearly arousal. In the Appendix, we consider the third principal component as well, but the picture is no clearer. Once again, this provides evidence against a simple two factor model of emotion. Below, we further investigate the valence/arousal model.

## 6.4 VALIDATION ON THE OPEN AFFECTIVE STANDARDIZED IMAGE SET (OASIS)

The Open Affective Standardized Image Set (OASIS) dataset (Kurdi et al., 2017) was developed as an alternative to the International Affective Picture System ((Lang, 2005)), a set of emotional stimuli which has been used in many psychology studies. OASIS consists of 900 color images which were rated by human judges on MTurk along the valence and arousal scales discussed above. OASIS images come in classes with labels such as "Alcohol", "Flowers", or "Pigeon." We applied our Deep Sentiment model to these images, treating the single-word label as the input text. After making predictions, we projected them onto the principal components discussed above.

In Table 4 we calculated the correlations between the first three principal components from our model and the mean valence and arousal variables from OASIS. The high correlation between the first principal component (PC1) and the valence variable provides evidence, as in the previous section, that PC1 is capturing a dimension of emotion separating positive from negative. It is also interesting to note the low correlation between PC1 and arousal. By contrast, while PC2 and PC3 are correlated with arousal, the correlations are not very high, and they are also correlated with valence. On the one hand, this is more evidence against there being two factors explaining emotions. On the other hand, asking judges to rate a set of images based on arousal and valence is more similar to the standard sentiment analysis task of predicting whether a sentence expresses positive or negative emotion. By contrast, our dataset comes from a set of images which were chosen by users who also decided to select emotion word tags to accompany them. We consider this to thus be a dataset of richer emotional content, and by design it represents a richer set of emotions.

Table 4: Correlation between OASIS arousal/valence and principal components

|  | OASIS valence | OASIS arousal |
|---|---|---|
| **PC1** | 58% | 3% |
| **PC2** | 17% | 22% |
| **PC3** | 30% | 11% |

## 7 CONCLUSION

We developed a novel multimodal sentiment analysis method using deep learning methods. Our goal was to investigate a core area of psychology, the study of emotion, using a large and novel social media dataset. Our approach provides new tools for the joint study of images and text on social media. This is important as social science researchers have begun to uncover the important role that images play on social media. As our dataset consisted of text, images, and emotion word tags, we considered it to be a form of "self-reported" data, which is the gold standard in emotion. However, it could also be considered to be an unobtrusive behavioral measure, and thus not subject to the biases inherent in laboratory studies. But because the data comes from public social media posts, we cannot rule out various sources of bias. What people choose to post—and not to post—on social media is influenced by how they want to present themselves (Wojnicki & Godes, 2008), so our study is limited insofar as it covers emotion not necessarily as it is truly experienced but rather as it is expressed or performed online.

In the future, we will evaluate our model on other image / text stimuli datasets that have been developed for psychological studies and investigate whether human judges are more or less accurate than our model. Finally, we will investigate other psychological components of the structure of emotion, for example daily and day of week trends in emotion.

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

## A   MORE EXAMPLES OF TUMBLR POSTS

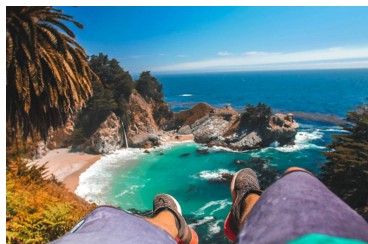

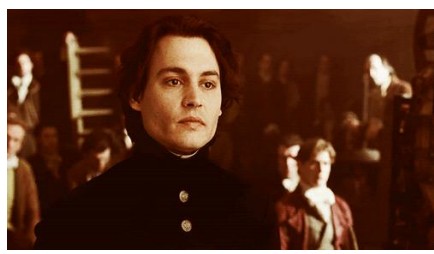

(a) **Happy**: "Just relax with this amazing view #bigsur #california #roadtrip #usa #life #fitness (at McWay Falls)"

(b) **Disgusted**: "Me when I see a couple expressing their affection in physical ways in public"

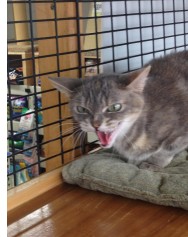

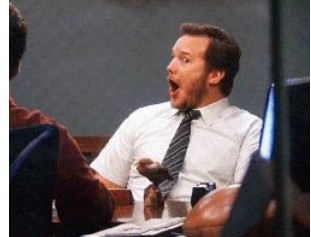

(c) **Angry**: "Tensions were high this Caturday..."

(d) **Surprised**: "Which Tea? Peppermint tea: What is your favorite gif right now?"

Figure 8: Tumblr posts

## B   FURTHER RESULTS

Table 5: Top 11-20 words for each emotion

| Emotion | Top words |
|---------|-----------|
| **Happy** | beautiful, beauty, fun, nice, dinner, sweet, good, loves, amazing, comfortable |
| **Calm** | view, sound, cool, continue, visit, push, woke, morning, safe, step |
| **Sad** | cry, sick, hurt, dead, feeling, worse, death, upset, falling, panic |
| **Scared** | kill, happened, cause, asleep, anxiety, worry, hurt, happen, worst, seriously |
| **Bored** | sitting, suddenly, freaking, slowly, annoying, sometimes, gotten, completely, older, cold |
| **Angry** | hurt, face, against, cause, hurts, fucking, fight, pain, strong, worst |
| **Annoyed** | depressed, lately, conversation, constantly, scared, stupid, scares, bored, heard, extremely |
| **Love** | sweet, mother, forget, hate, her, song, enjoy, sister, wonderful, dear |
| **Excited** | 2017, positive, am, definitely, awesome, happens, listening, happy, grow, wanna |
| **Surprised** | annoyed, negative, apparently, u, minute, asked, sadness, happened, moments, laugh |
| **Optimistic** | view, yet, believe, tomorrow, trending, said, situation, mood, forward, future |
| **Amazed** | im, quite, appreciate, honestly, knew, learned, yang, felt, liked, asleep |
| **Ashamed** | cried, afraid, annoyed, okay, sad, sick, pissed, fucking, depressed, wrong |
| **Disgusted** | tumblr, pissed, anger, sorry, cried, fucking, terrified, honestly, scared, facebook |
| **Pensive** | depression, text, voice, lonely, soul, by, read, truth, sounds, middle |

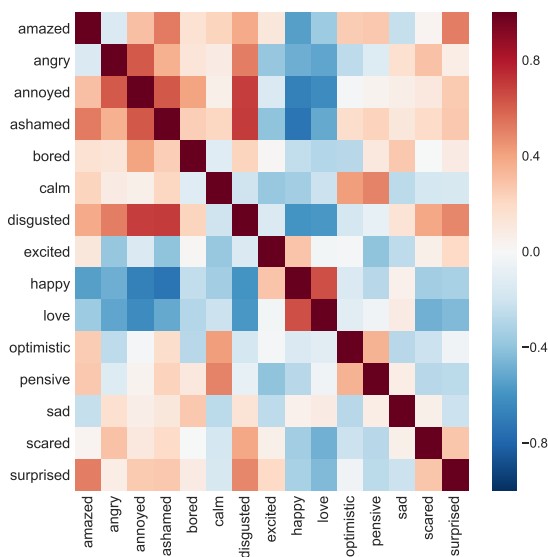

Figure 9: Heatmap of the emotion correlation matrix

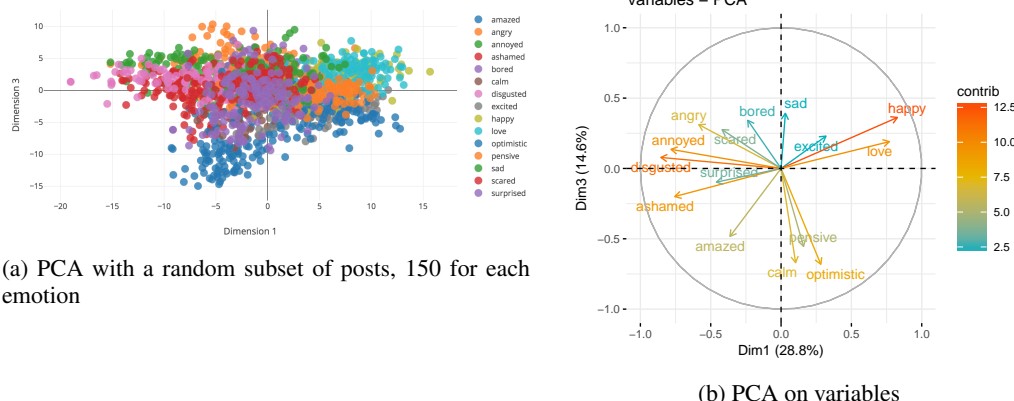

(a) PCA with a random subset of posts, 150 for each emotion

(b) PCA on variables

Figure 10: PCA on dimension 1 and 3

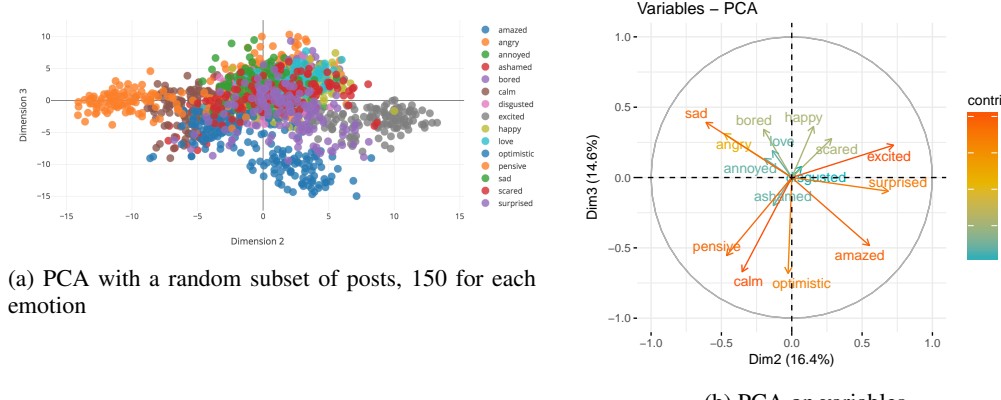

(a) PCA with a random subset of posts, 150 for each emotion

(b) PCA on variables

Figure 11: PCA for dimension 2 and 3

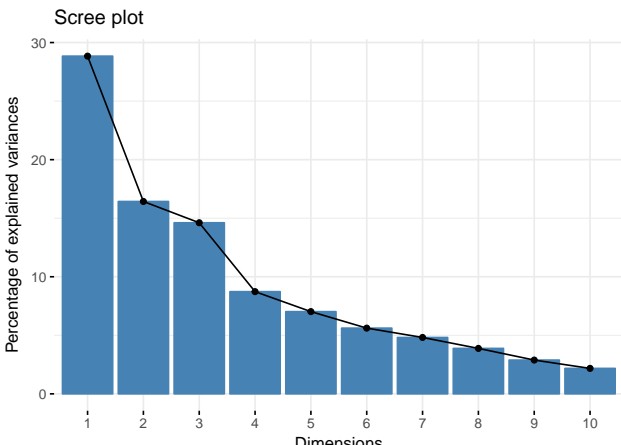

Figure 12: Explained variance of the PCA

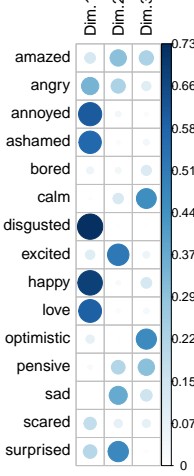

Figure 13: Factor loadings

