# OpenReview forum: "Multimodal Sentiment Analysis To Explore the Structure of Emotions"
_ICLR.cc/2018/Conference — Reject_

### Official Review · AnonReviewer2 · 2017-11-24
**The authors present a study that aims at inferring the "emotional" tags provided by Thumblr users starting from images and texts in the captions.**

**Rating:** 6
**Confidence:** 5

**Review:**


The authors present a study that aims at inferring the "emotional" tags provided by Thumblr users starting from images and texts in the captions. For text processing the authors use a standard LSTM taking as input GLOVE vectors of words in a sentence. For visual information, authors use a pretrained CNN (with fine tuning). A fully connected layer is used to fuse the multimodal information. Experimental results are reported in a self generated data set.

The contribution from the RL perspective is limited, in the sense that the authors simply applied standard models to predict a bunch of labels (in this case, emotion labels). It is interesting the "psychological" analysis that the authors present in Section 6. Still, I think the contribution in that part is a: sentiment-psychologically inspired analysis of the Thumbrl data set.

I think the author's statement on that this study leads to a more plausible psychological model of emotion is not well founded (they also mention to learn to recognize the latent emotional state). Whereas it is true that psychological studies rely on self - filled questionnaires, comparing a questionnaire (produced by expert psychologist) to the tags provided by users in a social network is to ambitious. (in some parts the authors make explicit this is an approximation, this should be stressed in every part of the paper)

---

> ### Author Response · Authors · 2018-01-05
> **Reply**
>
> Thank you for the review.
>
> “The contribution from the RL perspective is limited, in the sense that the authors simply applied standard models to predict a bunch of labels (in this case, emotion labels)”
> We wouldn’t qualify our model to just be predicting a “bunch of labels” given the complexity of inferring emotional states (due to the high intra class variability). The main contribution of the paper is that we investigate the study of emotion with a novel and large dataset including images (which is not as readily available on other social media such as Twitter), and further use the model to examine psychological components of the structure of emotion.

---

### Official Review · AnonReviewer3 · 2017-11-27
**Hashtag classification of Tumblr Posts**

**Rating:** 4
**Confidence:** 5

**Review:**

This paper presents a method for classifying Tumblr posts with associated images according to associated single emotion word hashtags.  The method relies on sentiment pre-processing from GloVe and image pre-processing from Inception.

My strongest criticism for this paper is against the claim that Tumblr post represent self-reported emotions and that this method sheds new insight on emotion representation and my secondary criticism is a lack of novelty in the method, which seems to be simply a combination of previously published sentiment analysis module and previously published image analysis module, fused in an output layer.

The authors claim that the hashtags represent self-reported emotions, but this is not true in the way that psychologists query participants regarding emotion words in psychology studies.  Instead these are emotion words that a person chooses to broadcast along with an associated announcement.  As the authors point out, hashtags and words may be used sarcastically or in different ways from what is understood in emotion theory.  It is quite common for everyday people to use emotion words this way e.g. using #love to express strong approval rather than an actual feeling of love.

In their analysis the authors claim:
“The 15 emotions retained were those with high relative frequencies on Tumblr among the PANAS-X scale (Watson & Clark, 1999)”.
However five of the words the authors retain: bored, annoyed, love, optimistic, and pensive are not in fact found in the PANAS-X scale:

Reference: The PANAS-X Scale: https://wiki.aalto.fi/download/attachments/50102838/PANAS-X-scale_spec.pdf Also the longer version that the authors cited:
https://www2.psychology.uiowa.edu/faculty/clark/panas-x.pdf

It should also be noted that the PANAS (Positive and Negative Affect Scale) scale and the PANAS-X (the “X” is for eXtended) scale are questionnaires used to elicit from participants feelings of positive and negative affect, they are not collections of "core" emotion words, but rather words that are colloquially attached to either positive or negative sentiment.  For example PANAS-X includes words like:“strong” ,“active”, “healthy”, “sleepy” which are not considered emotion words by psychology.

If the authors stated goal is "different than the standard sentiment analysis goal of predicting whether a sentence expresses positive or negative sentiment" they should be aware that this is exactly what PANAS is designed to do - not to infer the latent emotional state of a person, except to the extent that their affect is positive or negative.


The work of representing emotions had been an field in psychology for over a hundred years and it is still continuing.  https://en.wikipedia.org/wiki/Contrasting_and_categorization_of_emotions.

One of the most popular theories of emotion is the theory that there exist “basic” emotions: Anger, Disgust, Fear, Happiness (enjoyment), Sadness and Surprise (Paul Ekman, cited by the authors).  These are short duration sates lasting only seconds.  They are also fairly specific, for example “surprise” is sudden reaction to something unexpected, which is it exactly the same as seeing a flower on your car and expressing “what a nice surprise.”  The surprise would be the initial reaction of “what’s that on my car?  Is it dangerous?” but after identifying the object as non-threatening, the emotion of “surprise” would likely pass and be replaced with appreciation.

The Circumplex Model of Emotions (Posner et al 2005) the authors refer to actually stands in opposition to the theories of Ekman.  From the cited paper by Posner et al :
"The circumplex model of affect proposes that all affective states arise from cognitive interpretations of core neural sensations that are the product of two independent neurophysiological systems. This model stands in contrast to theories of basic emotions, which posit that a discrete and independent neural system subserves every emotion."
From my reading of this paper, it is clear to me that the authors do not have a clear understanding of the current state of psychology’s view of emotion representation and this work would not likely contribute to a new understanding of the latent structure of peoples’ emotions.

In the PCA result, it is not "clear" that the first axis represents valence, as "sad" has a slight positive on this scale and "sad" is one of the emotions most clearly associated with negative valence.

With respect to the rest of the paper, the level of novelty and impact is "ok, but not good enough."  This analysis does not seem very different from Twitter analysis, because although Tumblr posts are allowed to be longer than Twitter posts, the authors truncate the posts to 50 characters.  Additionally, the images do not seem to add very much to the classification.  The authors algorithm also seems to be essentially a combination of two other, previously published algorithms.

For me the novelty of this paper was in its application to the realm of emotion theory, but I do not feel there is a contribution here.  This paper is more about classifying Tumblr posts according to emotion word hashtags than a paper that generates a new insights into emotion representation or that can infer latent emotional state.

---

> ### Author Response · Authors · 2018-01-05
> **Reply (part 1)**
>
> 1) “The authors claim that the hashtags represent self-reported emotions, but this is not true in the way that psychologists query participants regarding emotion words in psychology studies.  Instead these are emotion words that a person chooses to broadcast along with an associated announcement.  As the authors point out, hashtags and words may be used sarcastically or in different ways from what is understood in emotion theory.  It is quite common for everyday people to use emotion words this way e.g. using #love to express strong approval rather than an actual feeling of love.”
>
> As we describe in the paper, there is no agreed upon gold-standard for measuring emotion in psychology. Self-report is considered the best, but there can be demand effects (Orne 1962) through which subjects try to tailor their responses in some way due to the fact that they are participating in a study. By contrast, behavioral measures (Webb et al, 1966) can be more reliable as they are less subject to demand effects. We see all of the performance by users on Tumblr as behavioral, with the emotion tags are as a behavioral report of emotion. We agree that there is noise inherent in this measure, due to, e.g. sarcasm, but did not see reason to worry that this was a significant source of bias.
>
> 2) “In their analysis the authors claim:
> “The 15 emotions retained were those with high relative frequencies on Tumblr among the PANAS-X scale (Watson & Clark, 1999)”.
> However five of the words the authors retain: bored, annoyed, love, optimistic, and pensive are not in fact found in the PANAS-X scale:
>
> Reference: The PANAS-X Scale: https://wiki.aalto.fi/download/attachments/50102838/PANAS-X-scale_spec.pdf Also the longer version that the authors cited:
> https://www2.psychology.uiowa.edu/faculty/clark/panas-x.pdf
>
> It should also be noted that the PANAS (Positive and Negative Affect Scale) scale and the PANAS-X (the “X” is for eXtended) scale are questionnaires used to elicit from participants feelings of positive and negative affect, they are not collections of "core" emotion words, but rather words that are colloquially attached to either positive or negative sentiment.  For example PANAS-X includes words like:“strong” ,“active”, “healthy”, “sleepy” which are not considered emotion words by psychology. ”
>
> Good point, the five emotions not appearing in the PANAS-X scale were found in the Plutchik's Wheel of Emotions. We extracted as many posts as possible for various emotions and kept the 15 emotions with the highest relative frequencies. We clarified that point in the paper, page 2.
>
> 3) “If the authors stated goal is "different than the standard sentiment analysis goal of predicting whether a sentence expresses positive or negative sentiment" they should be aware that this is exactly what PANAS is designed to do - not to infer the latent emotional state of a person, except to the extent that their affect is positive or negative.”
>
> By standard sentiment analysis, we simply mean methods designed to categorize a sentence like "I loved the new Star Wars movie!" as positive. Very simple methods (e.g. LIWC) can do a decent job at this task. But do these methods capture latent emotional state? (Whether this emotional state is conceived of as positive/negative affect, core emotions, or a circumplex model is a separate issue we discuss below). We argue that there is strong evidence that standard sentiment analysis methods do NOT correspond to the latent emotional of the user and have added a citation (Flaxman and Kassam, 2016) backing up this claim. This is what motivates our attempts to find a new sentiment analysis method.

---

> ### Author Response · Authors · 2018-01-05
> **Reply (part 2)**
>
> 4) “The work of representing emotions had been an field in psychology for over a hundred years and it is still continuing.  https://en.wikipedia.org/wiki/Contrasting_and_categorization_of_emotions.
>
> One of the most popular theories of emotion is the theory that there exist “basic” emotions: Anger, Disgust, Fear, Happiness (enjoyment), Sadness and Surprise (Paul Ekman, cited by the authors).  These are short duration sates lasting only seconds.  They are also fairly specific, for example “surprise” is sudden reaction to something unexpected, which is it exactly the same as seeing a flower on your car and expressing “what a nice surprise.”  The surprise would be the initial reaction of “what’s that on my car?  Is it dangerous?” but after identifying the object as non-threatening, the emotion of “surprise” would likely pass and be replaced with appreciation.
>
> The Circumplex Model of Emotions (Posner et al 2005) the authors refer to actually stands in opposition to the theories of Ekman.  From the cited paper by Posner et al :
> "The circumplex model of affect proposes that all affective states arise from cognitive interpretations of core neural sensations that are the product of two independent neurophysiological systems. This model stands in contrast to theories of basic emotions, which posit that a discrete and independent neural system subserves every emotion."
> From my reading of this paper, it is clear to me that the authors do not have a clear understanding of the current state of psychology’s view of emotion representation and this work would not likely contribute to a new understanding of the latent structure of peoples’ emotions.”
>
> We agree with your characterization of two of the theories in the literature, but as you say this work continues. As we believe this is by no means a settled field, we see our contribution as that of providing a new measurement tool, more robust than standard sentiment analysis, for the automatic measurement of emotion. We reference both Ekman and the Circumplex model because we see our work as attempting to find evidence, positive or negative, for these theories. But indeed, a longer a more detailed study is needed--we have just scratched the surface in terms of creating a relevant dataset and showing that simple neural network models can achieve good accuracy on this dataset.
>
> 5) “In the PCA result, it is not "clear" that the first axis represents valence, as "sad" has a slight positive on this scale and "sad" is one of the emotions most clearly associated with negative valence.”
>
> We agree and will update our text with caveats accordingly.
>
> 6) “With respect to the rest of the paper, the level of novelty and impact is "ok, but not good enough."  This analysis does not seem very different from Twitter analysis, because although Tumblr posts are allowed to be longer than Twitter posts, the authors truncate the posts to 50 characters.”
>
> We truncate to 50 words, not 50 characters, and therefore if we consider that on average an English word contains 4.5 characters (http://www.cs.trincoll.edu/~crypto/resources/LetFreq.html), the Tumblr text is 60% longer than the maximum Twitter post, and probably more than twice longer than the average Twitter post.
>
> 7) “Additionally, the images do not seem to add very much to the classification.  The authors algorithm also seems to be essentially a combination of two other, previously published algorithms.
>
> For me the novelty of this paper was in its application to the realm of emotion theory, but I do not feel there is a contribution here.  This paper is more about classifying Tumblr posts according to emotion word hashtags than a paper that generates a new insights into emotion representation or that can infer latent emotional state.”
>
> Thank you for your careful reading of our paper.

---

### Official Review · AnonReviewer1 · 2017-11-27
**Interesting paper on sentiment analysis combining image and text**

**Rating:** 5
**Confidence:** 5

**Review:**

The paper presents a multi-modal CNN model for sentiment analysis that combines images and text.  The model is trained on a new dataset collected from Tumblr.

Positive aspects:
+ Emphasis in model interpretability and its connection to psychological findings in emotions
+ The idea of using Tumblr data seems interesting, allowing to work with a large set of emotion categories, instead of considering just the binary task positive vs. negative.

Weaknesses:
- A deeper analysis of previous work on the combination of image and text for sentiment analysis (both datasets and methods) and its relation with the presented work is necessary.
- The proposed method is not compared with other methods that combine text and image for sentiment analysis.
-  The study is limited to just one dataset.

The paper presents interesting ideas and findings in an important challenging area. The main novelties of the paper are: (1) the use of Tumblr data, (2) the proposed CNN architecture, combining images and text (using word embedding.

I missed a "related work section", where authors clearly mention previous works on similar datasets. Some related works are mentioned in the paper, but those are spread in different sections. It's hard to get a clear overview of the previous research: datasets, methods and contextualization of the proposed approach in relation with previous work. I think authors should cite Sentibanks. Also, at some point authors should compare their proposal with previous work.

More comments:

- Some figures could be more complete: to see more examples in Fig 1, 2, 3 would help to understand better the dataset and the challenges.
- In table 4, for example, it would be nice to see the performance on the different emotion categories.
- It would be interesting to see qualitative visual results on recognitions.

I like this work, but I think authors should improve the aspects I mention for its publication.

---

> ### Author Response · Authors · 2018-01-05
> **Reply**
>
> Thank you very much for your comments which helped us restructure the paper that is hopefully more intelligible now.
>
> “A deeper analysis of previous work on the combination of image and text for sentiment analysis (both datasets and methods) and its relation with the presented work is necessary.”
> We added a “Related work” section (page 2) to better contextualise our proposed model with what has been previously done in visual and textual sentiment analysis.
>
> “Some figures could be more complete: to see more examples in Fig 1, 2, 3 would help to understand better the dataset and the challenges.”
> More examples of Tumblr posts are now in the Appendix, page 13.

---

### Decision · Program_Chairs · 2018-01-29
**ICLR 2018 Conference Acceptance Decision**

**Decision:**

Reject

**Comment:**

This work combines words and images from Tumblr to provide more fine-grained sentiment analysis than just positive-negative. The contribution is too slight, as a straightforward combination of existing architectures applied on an emotion classification task with conclusions that aren't well motivated and are not providing any comparison to existing related work on finer emotion classification.